# Fair Clustering Through Fairlets

**Flavio Chierichetti**
Dipartimento di Informatica
Sapienza University
Rome, Italy

**Ravi Kumar**
Google Research
1600 Amphitheater Parkway
Mountain View, CA 94043

**Silvio Lattanzi**
Google Research
76 9th Ave
New York, NY 10011

**Sergei Vassilvitskii**
Google Research
76 9th Ave
New York, NY 10011

## Abstract

We study the question of fair clustering under the *disparate impact* doctrine, where each protected class must have approximately equal representation in every cluster. We formulate the fair clustering problem under both the $k$-center and the $k$-median objectives, and show that even with two protected classes the problem is challenging, as the optimum solution can violate common conventions—for instance a point may no longer be assigned to its nearest cluster center!

En route we introduce the concept of *fairlets*, which are minimal sets that satisfy fair representation while approximately preserving the clustering objective. We show that any fair clustering problem can be decomposed into first finding good fairlets, and then using existing machinery for traditional clustering algorithms. While finding good fairlets can be NP-hard, we proceed to obtain efficient approximation algorithms based on minimum cost flow.

We empirically demonstrate the *price of fairness* by quantifying the value of fair clustering on real-world datasets with sensitive attributes.

## 1 Introduction

From self driving cars, to smart thermostats, and digital assistants, machine learning is behind many of the technologies we use and rely on every day. Machine learning is also increasingly used to aid with decision making—in awarding home loans or in sentencing recommendations in courts of law (Kleinberg *et al.* , 2017a). While the learning algorithms are not inherently biased, or unfair, the algorithms may pick up and amplify biases already present in the training data that is available to them. Thus a recent line of work has emerged on designing *fair* algorithms.

The first challenge is to formally define the concept of fairness, and indeed recent work shows that some natural conditions for fairness cannot be simultaneously achieved (Kleinberg *et al.* , 2017b; Corbett-Davies *et al.* , 2017). In our work we follow the notion of *disparate impact* as articulated by Feldman *et al.* (2015), following the *Griggs v. Duke Power Co.* US Supreme Court case. Informally, the doctrine codifies the notion that not only should protected attributes, such as race and gender, not be *explicitly* used in making decisions, but even after the decisions are made they should not be disproportionately different for applicants in different protected classes. In other words, if an unprotected feature, for example, height, is closely correlated with a protected feature, such as gender, then decisions made based on height may still be unfair, as they can be used to effectively discriminate based on gender.

While much of the previous work deals with supervised learning, in this work we consider the most common unsupervised learning problem, that of clustering. In modern machine learning systems, clustering is often used for feature engineering, for instance augmenting each example in the dataset with the id of the cluster it belongs to in an effort to bring expressive power to simple learning methods. In this way we want to make sure that the features that are generated are fair themselves. As in standard clustering literature, we are given a set $X$ of points lying in some metric space, and our goal is to find a partition of $X$ into $k$ different clusters, optimizing a particular objective function. We assume that the coordinates of each point $x \in X$ are unprotected; however each point also has a color, which identifies its protected class. The notion of disparate impact and fair representation then translates to that of color balance in each cluster. We study the two color case, where each point is either *red* or *blue*, and show that even this simple version has a lot of underlying complexity. We formalize these views and define a fair clustering objective that incorporates both fair representation and the traditional clustering cost; see Section 2 for exact definitions.

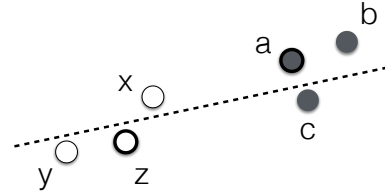

Figure 1: A colorblind $k$-center clustering algorithm would group points $a, b, c$ into one cluster, and $x, y, z$ into a second cluster, with centers at $a$ and $z$ respectively. A fair clustering algorithm, on the other hand, may give a partition indicated by the dashed line. Observe that in this case a point is no longer assigned to its nearest cluster center. For example $x$ is assigned to the same cluster as $a$ even though $z$ is closer.

A clustering algorithm that is *colorblind*, and thus does not take a protected attribute into its decision making, may still result in very unfair clusterings; see Figure 1. This means that we must explicitly use the protected attribute to find a fair solution. Moreover, this implies that a fair clustering solution could be strictly worse (with respect to an objective function) than a colorblind solution.

Finally, the example in Figure 1 also shows the main technical hurdle in looking for fair clusterings. Unlike the classical formulation where every point is assigned to its nearest cluster center, this may no longer be the case. Indeed, a fair clustering is defined not just by the position of the centers, but also by an *assignment function* that assigns a cluster label to each input.

**Our contributions.** In this work we show how to reduce the problem of fair clustering to that of classical clustering via a pre-processing step that ensures that any resulting solution will be fair. In this way, our approach is similar to that of Zemel *et al.* (2013), although we formulate the first step as an explicit combinatorial problem, and show approximation guarantees that translate to approximation guarantees on the optimal solution. Specifically we:

(i) Define fair variants of classical clustering problems such as $k$-center and $k$-median;
(ii) Define the concepts of fairlets and fairlet decompositions, which encapsulate minimal fair sets;
(iii) Show that any fair clustering problem can be reduced to first finding a fairlet decomposition, and then using the classical (not necessarily fair) clustering algorithm;
(iv) Develop approximation algorithms for finding fair decompositions for a large range of fairness values, and complement these results with NP-hardness; and
(v) Empirically quantify the price of fairness, i.e., the ratio of the cost of traditional clustering to the cost of fair clustering.

**Related work.** Data clustering is a classic problem in unsupervised learning that takes on many forms, from partition clustering, to soft clustering, hierarchical clustering, spectral clustering, among many others. See, for example, the books by Aggarwal & Reddy (2013); Xu & Wunsch (2009) for an extensive list of problems and algorithms. In this work, we focus our attention on the $k$-center and $k$-median problems. Both of these problems are NP-hard but have known efficient approximation algorithms. The state of the art approaches give a 2-approximation for $k$-center (Gonzalez, 1985) and a $(1 + \sqrt{3} + \epsilon)$-approximation for $k$-median (Li & Svensson, 2013).

Unlike clustering, the exploration of fairness in machine learning is relatively nascent. There are two broad lines of work. The first is in codifying what it means for an algorithm to be fair. See for example the work on statistical parity (Luong *et al.* , 2011; Kamishima *et al.* , 2012), disparate impact (Feldman *et al.* , 2015), and individual fairness (Dwork *et al.* , 2012). More recent work

by Corbett-Davies *et al.* (2017) and Kleinberg *et al.* (2017b) also shows that some of the desired properties of fairness may be incompatible with each other.

A second line of work takes a specific notion of fairness and looks for algorithms that achieve fair outcomes. Here the focus has largely been on supervised learning (Luong *et al.* , 2011; Hardt *et al.* , 2016) and online (Joseph *et al.* , 2016) learning. The direction that is most similar to our work is that of learning intermediate representations that are guaranteed to be fair, see for example the work by Zemel *et al.* (2013) and Kamishima *et al.* (2012). However, unlike their work, we give strong guarantees on the relationship between the quality of the fairlet representation, and the quality of any fair clustering solution.

In this paper we use the notion of fairness known as *disparate impact* and introduced by Feldman *et al.* (2015). This notion is also closely related to the $p\%$-rule as a measure for fairness. The $p\%$-rule is a generalization of the $80\%$-rule advocated by US Equal Employment Opportunity Commission (Biddle, 2006) and was used in a recent paper on mechanism for fair classification (Zafar *et al.* , 2017). In particular our paper addresses an open question of Zafar *et al.* (2017) presenting a framework to solve an unsupervised learning task respecting the $p\%$-rule.

## 2 Preliminaries

Let $X$ be a set of points in a metric space equipped with a distance function $d : X^2 \to \mathbb{R}^{\geq 0}$. For an integer $k$, let $[k]$ denote the set $\{1, \ldots, k\}$.

We first recall standard concepts in clustering. A *k-clustering* $\mathcal{C}$ is a partition of $X$ into $k$ disjoint subsets, $C_1, \ldots, C_k$, called *clusters*. We can evaluate the quality of a clustering $\mathcal{C}$ with different objective functions. In the *k-center* problem, the goal is to minimize
$$\phi(X, \mathcal{C}) = \max_{C \in \mathcal{C}} \min_{c \in C} \max_{x \in C} d(x, c),$$
and in the *k-median* problem, the goal is to minimize
$$\psi(X, \mathcal{C}) = \sum_{C \in \mathcal{C}} \min_{c \in C} \sum_{x \in C} d(x, c).$$
A clustering $\mathcal{C}$ can be equivalently described via an *assignment* function $\alpha : X \to [k]$. The points in cluster $C_i$ are simply the pre-image of $i$ under $\alpha$, i.e., $C_i = \{x \in X \mid \alpha(x) = i\}$.

Throughout this paper we assume that each point in $X$ is colored either red or blue; let $\chi : X \to \{\text{RED}, \text{BLUE}\}$ denote the color of a point. For a subset $Y \subseteq X$ and for $c \in \{\text{RED}, \text{BLUE}\}$, let $c(Y) = \{x \in X \mid \chi(x) = c\}$ and let $\#c(Y) = |c(Y)|$.

We first define a natural notion of balance.

**Definition 1** (Balance). *For a subset $\varnothing \neq Y \subseteq X$, the* balance *of $Y$ is defined as:*
$$\text{balance}(Y) = \min \left( \frac{\#\text{RED}(Y)}{\#\text{BLUE}(Y)}, \frac{\#\text{BLUE}(Y)}{\#\text{RED}(Y)} \right) \in [0, 1].$$
*The balance of a clustering $\mathcal{C}$ is defined as:*
$$\text{balance}(\mathcal{C}) = \min_{C \in \mathcal{C}} \text{balance}(C).$$

A subset with an equal number of red and blue points has balance 1 (perfectly balanced) and a monochromatic subset has balance 0 (fully unbalanced). To gain more intuition about the notion of balance, we investigate some basic properties that follow from its definition.

**Lemma 2** (Combination). *Let $Y, Y' \subseteq X$ be disjoint. If $\mathcal{C}$ is a clustering of $Y$ and $\mathcal{C}'$ is a clustering of $Y'$, then* $\text{balance}(\mathcal{C} \cup \mathcal{C}') = \min(\text{balance}(\mathcal{C}), \text{balance}(\mathcal{C}'))$.

It is easy to see that for any clustering $\mathcal{C}$ of $X$, we have $\text{balance}(\mathcal{C}) \leq \text{balance}(X)$. In particular, if $X$ is not perfectly balanced, then no clustering of $X$ can be perfectly balanced. We next show an interesting converse, relating the balance of $X$ to the balance of a well-chosen clustering.

**Lemma 3.** *Let $\text{balance}(X) = b/r$ for some integers $1 \leq b \leq r$ such that $\gcd(b, r) = 1$. Then there exists a clustering $\mathcal{Y} = \{Y_1, \ldots, Y_m\}$ of $X$ such that (i) $|Y_j| \leq b + r$ for each $Y_j \in \mathcal{Y}$, i.e., each cluster is small, and (ii) $\text{balance}(\mathcal{Y}) = b/r = \text{balance}(X)$.*

**Fairness and fairlets.** Balance encapsulates a specific notion of fairness, where a clustering with a monochromatic cluster (i.e., fully unbalanced) is considered unfair. We call the clustering $\mathcal{Y}$ as described in Lemma 3 a $(b, r)$-*fairlet decomposition* of $X$ and call each cluster $Y \in \mathcal{Y}$ a *fairlet*.

Equipped with the notion of balance, we now revisit the clustering objectives defined earlier. The objectives do not consider the color of the points, so they can lead to solutions with monochromatic clusters. We now extend them to incorporate fairness.

**Definition 4** (($(t, k)$-fair clustering problems). *In the $(t, k)$-fair center (resp., $(t, k)$-fair median) problem, the goal is to partition $X$ into $\mathcal{C}$ such that $|\mathcal{C}| = k$, $\mathrm{balance}(\mathcal{C}) \geq t$, and $\phi(X, \mathcal{C})$ (resp. $\psi(X, \mathcal{C})$) is minimized.*

Traditional formulations of $k$-center and $k$-median eschew the notion of an assignment function. Instead it is implicit through a set $\{c_1, \dots, c_k\}$ of centers, where each point assigned to its nearest center, i.e., $\alpha(x) = \arg\min_{i \in [1,k]} d(x, c_i)$. Without fairness as an issue, they are equivalent formulations; however, with fairness, we need an explicit assignment function (see Figure 1).

Missing proofs are deferred to the full version of the paper.

# 3  Fairlet decomposition and fair clustering

At first glance, the fair version of a clustering problem appears harder than its vanilla counterpart. In this section we prove, interestingly, a reduction from the former to the latter. We do this by first clustering the original points into small clusters preserving the balance, and then applying vanilla clustering on these smaller clusters instead of on the original points.

As noted earlier, there are different ways to partition the input to obtain a fairlet decomposition. We will show next that the choice of the partition directly impacts the approximation guarantees of the final clustering algorithm.

Before proving our reduction we need to introduce some additional notation. Let $\mathcal{Y} = \{Y_1, \dots, Y_m\}$ be a fairlet decomposition. For each cluster $Y_j$, we designate an arbitrary point $y_j \in Y_j$ as its *center*. Then for a point $x$, we let $\beta : X \to [1, m]$ denote the index of the fairlet to which it is mapped. We are now ready to define the cost of a fairlet decomposition

**Definition 5** (Fairlet decomposition cost). *For a fairlet decomposition, we define its $k$-median cost as $\sum_{x \in X} d(x, \beta(x))$, and its $k$-center cost as $\max_{x \in X} d(x, \beta(x))$. We say that a $(b, r)$-fairlet decomposition is* optimal *if it has minimum cost among all $(b, r)$-fairlet decompositions.*

Since $(X, d)$ is a metric, we have from the triangle inequality that for any other point $c \in X$,
$$d(x, c) \leq d(x, y_{\beta(x)}) + d(y_{\beta(x)}, c).$$

Now suppose that we aim to obtain a $(t, k)$-fair clustering of the original points $X$. (As we observed earlier, necessarily $t \leq \mathrm{balance}(X)$.) To solve the problem we can cluster instead the centers of each fairlet, i.e., the set $\{y_1, \dots, y_m\} = Y$, into $k$ clusters. In this way we obtain a set of centers $\{c_1, \dots, c_k\}$ and an assignment function $\alpha_Y : Y \to [k]$.

We can then define the overall assignment function as $\alpha(x) = \alpha_Y(y_{\beta(x)})$ and denote the clustering induced by $\alpha$ as $\mathcal{C}_\alpha$. From the definition of $\mathcal{Y}$ and the property of fairlets and balance, we get that $\mathrm{balance}(\mathcal{C}_\alpha) = t$. We now need to bound its cost. Let $\tilde{Y}$ be a multiset, where each $y_i$ appears $|Y_i|$ number of times.

**Lemma 6.** $\psi(X, \mathcal{C}_\alpha) = \psi(X, \mathcal{Y}) + \psi(\tilde{Y}, \mathcal{C}_\alpha)$ *and* $\phi(X, \mathcal{C}_\alpha) = \phi(X, \mathcal{Y}) + \phi(\tilde{Y}, \mathcal{C}_\alpha)$.

Therefore in both cases we can reduce the fair clustering problem to the problem of finding a good fairlet decomposition and then solving the vanilla clustering problem on the centers of the fairlets. We refer to $\psi(X, \mathcal{Y})$ and $\phi(X, \mathcal{Y})$ as the $k$-median and $k$-center costs of the fairlet decomposition.

# 4  Algorithms

In the previous section we presented a reduction from the fair clustering problem to the regular counterpart. In this section we use it to design efficient algorithms for fair clustering.

We first focus on the $k$-center objective and show in Section 4.3 how to adapt the reasoning to solve the $k$-median objective. We begin with the most natural case in which we require the clusters to be perfectly balanced, and give efficient algorithms for the $(1, k)$-fair center problem. Then we analyze the more challenging $(t, k)$-fair center problem for $t < 1$. Let $B = \textsc{blue}(X), R = \textsc{red}(X)$.

## 4.1 Fair $k$-center warmup: $(1,1)$-fairlets

Suppose $\text{balance}(X) = 1$, i.e., $(|R| = |B|)$ and we wish to find a perfectly balanced clustering. We now show how we can obtain it using a good $(1,1)$-fairlet decomposition.

**Lemma 7.** *An optimal $(1,1)$-fairlet decomposition for $k$-center can be found in polynomial time.*

*Proof.* To find the best decomposition, we first relate this question to a graph covering problem. Consider a bipartite graph $G = (B \cup R, E)$ where we create an edge $E = (b_i, r_j)$ with weight $w_{ij} = d(r_i, b_j)$ between any bichromatic pair of nodes. In this case a decomposition into fairlets corresponds to some perfect matching in the graph. Each edge in the matching represents a fairlet, $Y_i$. Let $\mathcal{Y} = \{Y_i\}$ be the set of edges in the matching.

Observe that the $k$-center cost $\phi(X, \mathcal{Y})$ is exactly the cost of the maximum weight edge in the matching, therefore our goal is to find a perfect matching that minimizes the weight of the maximum edge. This can be done by defining a threshold graph $G_\tau$ that has the same nodes as $G$ but only those edges of weight at most $\tau$. We then look for the minimum $\tau$ where the corresponding graph has a perfect matching, which can be done by (binary) searching through the $O(n^2)$ values.

Finally, for each fairlet (edge) $Y_i$ we can arbitrarily set one of the two nodes as the center, $y_i$. $\qquad\square$

Since any fair solution to the clustering problem induces a set of minimal fairlets (as described in Lemma 3), the cost of the fairlet decomposition found is at most the cost of the clustering solution.

**Lemma 8.** *Let $\mathcal{Y}$ be the partition found above, and let $\phi_t^*$ be the cost of the optimal $(t,k)$-fair center clustering. Then $\phi(X, \mathcal{Y}) \leq \phi_t^*$.*

This, combined with the fact that the best approximation algorithm for $k$-center yields a 2-approximation (Gonzalez, 1985) gives us the following.

**Theorem 9.** *The algorithm that first finds fairlets and then clusters them is a 3-approximation for the $(1,k)$-fair center problem.*

## 4.2 Fair $k$-center: $(1, t')$-fairlets

Now, suppose that instead we look for a clustering with balance $t \lesssim 1$. In this section we assume $t = 1/t'$ for some integer $t' > 1$. We show how to extend the intuition in the matching construction above to find approximately optimal $(1, t')$-fairlet decompositions for integral $t' > 1$.

In this case, we transform the problem into a *minimum cost flow* (MCF) problem.[1] Let $\tau > 0$ be a parameter of the algorithm. Given the points $B, R$, and an integer $t'$, we construct a directed graph $H_\tau = (V, E)$. Its node set $V$ is composed of two special nodes $\beta$ and $\rho$, all of the nodes in $B \cup R$, and $t'$ additional copies for each node $v \in B \cup R$. More formally,

$$V = \{\beta, \rho\} \cup B \cup R \cup \left\{ b_i^j \mid b_i \in B \text{ and } j \in [t'] \right\} \cup \left\{ r_i^j \mid r_i \in R \text{ and } j \in [t'] \right\}.$$

The directed edges of $H_\tau$ are as follows:
(i) A $(\beta, \rho)$ edge with cost 0 and capacity $\min(|B|, |R|)$.
(ii) A $(\beta, b_i)$ edge for each $b_i \in B$, and an $(r_i, \rho)$ edge for each $r_i \in R$. All of these edges have cost 0 and capacity $t' - 1$.
(iii) For each $b_i \in B$ and for each $j \in [t']$, a $(b_i, b_i^j)$ edge, and for each $r_i \in R$ and for each $j \in [t']$, an $(r_i, r_i^j)$ edge. All of these edges have cost 0 and capacity 1.
(iv) Finally, for each $b_i \in B, r_j \in R$ and for each $1 \leq k, \ell \leq t$, a $(b_i^k, r_j^\ell)$ edge with capacity 1. The cost of this edge is 1 if $d(b_i, r_j) \leq \tau$ and $\infty$ otherwise.

To finish the description of this MCF instance, we have now specify supply and demand at every node. Each node in $B$ has a supply of 1, each node in $R$ has a demand of 1, $\beta$ has a supply of $|R|$, and $\rho$ has a demand of $|B|$. Every other node has zero supply and demand. In Figure 2 we show an example of this construction for a small graph.

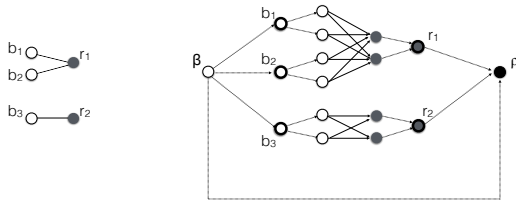

Figure 2: The construction of the MCF instance for the bipartite graph for $t' = 2$. Note that the only nodes with positive demands or supplies are $\beta, \rho, b_1, b_2, b_3, r_1$, and $r_2$ and all the dotted edges have cost 0.

The MCF problem can be solved in polynomial time and since all of the demands and capacities are integral, there exists an optimal solution that sends integral flow on each edge. In our case, the solution is a set of edges of $H_\tau$ that have non-zero flow, and the total flow on the $(\beta, \rho)$ edge.

In the rest of this section we assume for simplicity that any two distinct elements of the metric are at a positive distance apart and we show that starting from a solution to the described MCF instance we can build a low cost $(1, t')$-fairlet decomposition. We start by showing that every $(1, t')$-fairlet decomposition can be used to construct a feasible solution for the MCF instance and then prove that an optimal solution for the MCF instance can be used to obtain a $(1, t')$-fairlet decomposition.

**Lemma 10.** *Let $\mathcal{Y}$ be a $(1, t')$-fairlet decomposition of cost $C$ for the $(1/t', k)$-fair center problem. Then it is possible to construct a feasible solution of cost $2C$ to the MCF instance.*

*Proof.* We begin by building a feasible solution and then bound its cost. Consider each fairlet in the $(1, t')$-fairlet decomposition.

Suppose the fairlet contains 1 red node and $c$ blue nodes, with $c \le t'$, i.e., the fairlet is of the form $\{r_1, b_1, \ldots, b_c\}$. For any such fairlet we send a unit of flow form each node $b_i$ to $b_i^1$, for $i \in [c]$ and a unit of flow from nodes $b_1^1, \ldots, b_c^1$ to nodes $r_1^1, \ldots, r_1^c$. Furthermore we send a unit of flow from each $r_1^1, \ldots, r_1^c$ to $r_1$ and $c - 1$ units of flow from $r_1$ to $\rho$. Note that in this way we saturate the demands of all nodes in this fairlet.

Similarly, if the fairlet contains $c$ red nodes and 1 blue node, with $c \le t'$, i.e., the fairlet is of the form $\{r_1, \ldots, r_c, b_1\}$. For any such fairlet, we send $c - 1$ units of flow from $\beta$ to $b_1$. Then we send a unit of flow from each $b_1$ to each $b_1^1, \ldots, b_1^c$ and a unit of flow from nodes $b_1^1, \ldots, b_1^c$ to nodes $r_1^1, \ldots, r_c^1$. Furthermore we send a unit of flow from each $r_1^1, \ldots, r_c^1$ to the nodes $r_1, \ldots, r_c$. Note that also in this case we saturate all the request of nodes in this fairlet.

Since every node $v \in B \cup R$ is contained in a fairlet, all of the demands of these nodes are satisfied. Hence, the only nodes that can have still unsatisfied demand are $\beta$ and $\rho$, but we can use the direct edge $(\beta, \rho)$ to route the excess demand, since the total demand is equal to the total supply. In this way we obtain a feasible solution for the MCF instance starting from a $(1, t')$-fairlet decomposition.

To bound the cost of the solution note that the only edges with positive cost in the constructed solution are the edges between nodes $b_i^j$ and $r_k^\ell$. Furthermore an edge is part of the solution only if the nodes $b_i$ and $r_k$ are contained in the same fairlet $F$. Given that the $k$-center cost for the fairlet decomposition is $C$, the cost of the edges between nodes in $F$ in the constructed feasible solution for the MCF instance is at most 2 times this distance. The claim follows. $\square$

Now we show that given an optimal solution for the MCF instance of cost $C$, we can construct a $(1, t')$-fairlet decomposition of cost no bigger than $C$.

**Lemma 11.** *Let $\mathcal{Y}$ be an optimal solution of cost $C$ to the MCF instance. Then it is possible to construct a $(1, t')$-fairlet decomposition for $(1/t', k)$-fair center problem of cost at most $C$.*

Combining Lemma 10 and Lemma 11 yields the following.

**Lemma 12.** *By reducing the $(1, t')$-fairlet decomposition problem to an MCF problem, it is possible to compute a 2-approximation for the optimal $(1, t')$-fairlet decomposition for the $(1/t', k)$-fair center problem.*

Note that the cost of a $(1, t')$-fairlet decomposition is necessarily smaller than the cost of a $(1/t', k)$-fair clustering. Our main theorem follows.

**Theorem 13.** *The algorithm that first finds fairlets and then clusters them is a 4-approximation for the $(1/t', k)$-fair center problem for any positive integer $t'$.*

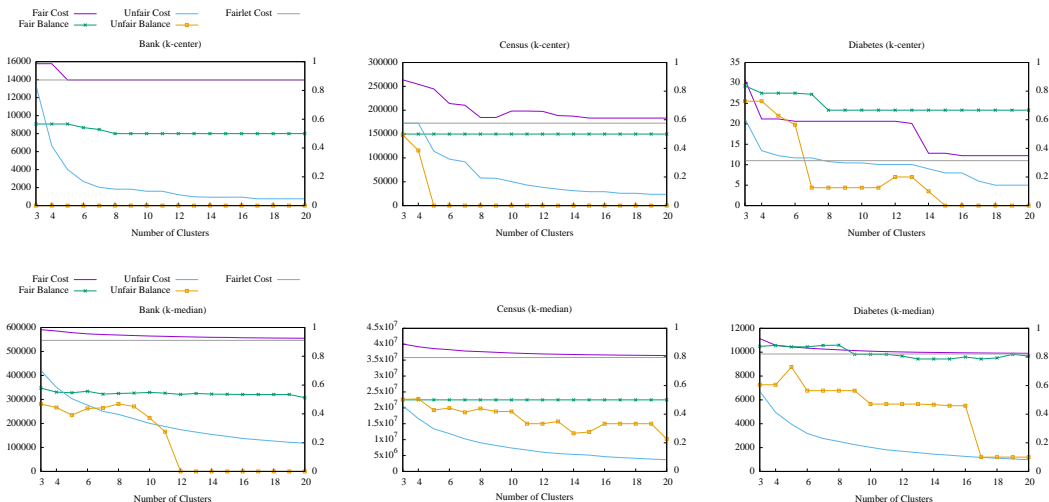

Figure 3: Empirical performance of the classical and fair clustering median and center algorithms on the three datasets. The cost of each solution is on left axis, and its balance on the right axis.

## 4.3 Fair $k$-median

The results in the previous section can be modified to yield results for the $(t, k)$-fair median problem with minor changes that we describe below.

For the perfectly balanced case, as before, we look for a perfect matching on the bichromatic graph. Unlike, the $k$-center case, we let the weight of a $(b_i, r_j)$ edge be the distance between the two points. Our goal is to find a perfect matching of minimum total cost, since that exactly represents the cost of the fairlet decomposition. Since the best known approximation for $k$-median is $1 + \sqrt{3} + \epsilon$ (Li & Svensson, 2013), we have:

**Theorem 14.** *The algorithm that first finds fairlets and then clusters them is a $(2 + \sqrt{3} + \epsilon)$-approximation for the $(1, k)$-fair median problem.*

To find $(1, t')$-fairlet decompositions for integral $t' > 1$, we again resort to MCF and create an instance as in the $k$-center case, but for each $b_i \in B, r_j \in R$, and for each $1 \leq k, \ell \leq t$, we set the cost of the edge $(b_i^k, r_j^\ell)$ to $d(b_i, r_j)$.

**Theorem 15.** *The algorithm that first finds fairlets and then clusters them is a $(t' + 1 + \sqrt{3} + \epsilon)$-approximation for the $(1/t', k)$-fair median problem for any positive integer $t'$.*

## 4.4 Hardness

We complement our algorithmic results with discussion of computational hardness for fair clustering. We show that the question of finding a good fairlet decomposition is itself computationally hard. Thus, ensuring fairness causes hardness, regardless of the underlying clustering objective.

**Theorem 16.** *For each fixed $t' \geq 3$, finding an optimal $(1, t')$-fairlet decomposition is NP-hard. Also, finding the minimum cost $(1/t', k)$-fair median clustering is NP-hard.*

# 5 Experiments

In this section we illustrate our algorithm by performing experiments on real data. The goal of our experiments is two-fold: first, we show that traditional algorithms for $k$-center and $k$-median tend to produce unfair clusters; second, we show that by using our algorithms one can obtain clusters that respect the fairness guarantees. We show that in the latter case, the cost of the solution tends to converge to the cost of the fairlet decomposition, which serves as a lower bound on the cost of the optimal solution.

**Datasets.** We consider 3 datasets from the UCI repository Lichman (2013) for experimentation.

*Diabetes.* This dataset[2] represents the outcomes of patients pertaining to diabetes. We chose numeric attributes such as age, time in hospital, to represent points in the Euclidean space and gender as the sensitive dimension, i.e., we aim to balance gender. We subsampled the dataset to 1000 records.

*Bank.* This dataset[3] contains one record for each phone call in a marketing campaign ran by a Portuguese banking institution (Moro *et al.* , 2014)). Each record contains information about the client that was contacted by the institution. We chose numeric attributes such as age, balance, and duration to represents points in the Euclidean space, we aim to cluster to balance married and not married clients. We subsampled the dataset to 1000 records.

*Census.* This dataset[4] contains the census records extracted from the 1994 US census (Kohavi, 1996). Each record contains information about individuals including education, occupation, hours worked per week, etc. . We chose numeric attributes such as age, fnlwgt, education-num, capital-gain and hours-per-week to represents points in the Euclidean space and we aim to cluster the dataset so to balance gender. We subsampled the dataset to 600 records.

**Algorithms.** We implement the flow-based fairlet decomposition algorithm as described in Section 4. To solve the $k$-center problem we augment it with the greedy furthest point algorithm due to Gonzalez (1985), which is known to obtain a 2-approximation. To solve the $k$-median problem we use the single swap algorithm due to Arya *et al.* (2004), which also gets a 5-approximation in the worst case, but performs much better in practice (Kanungo *et al.* , 2002).

**Results.** Figure 3 shows the results for $k$-center for the three datasets, and Figure 3 shows the same for the $k$-median objective. In all of the cases, we run with $t' = 2$, that is we aim for balance of at least $0.5$ in each cluster.

Observe that the balance of the solutions produced by the classical algorithms is very low, and in four out of the six cases, the balance is $0$ for larger values of $k$, meaning that the optimal solution has monochromatic clusters. Moreover, this is not an isolated incident, for instance the $k$-median instance of the Bank dataset has three monochromatic clusters starting at $k = 12$. Finally, left unchecked, the balance in all datasets keeps decreasing as the clustering becomes more discriminative, with increased $k$.

On the other hand the fair clustering solutions maintain a balanced solution even as $k$ increases. Not surprisingly, the balance comes with a corresponding increase in cost, and the fair solutions are costlier than their unfair counterparts. In each plot we also show the cost of the fairlet decomposition, which represents the limit of the cost of the fair clustering; in all of the scenarios the overall cost of the clustering converges to the cost of the fairlet decomposition.

# 6   Conclusions

In this work we initiate the study of fair clustering algorithms. Our main result is a reduction of fair clustering to classical clustering via the notion of fairlets. We gave efficient approximation algorithms for finding fairlet decompositions, and proved lower bounds showing that fairness can introduce a computational bottleneck. An immediate future direction is to tighten the gap between lower and upper bounds by improving the approximation ratio of the decomposition algorithms, or giving stronger hardness results. A different avenue is to extend these results to situations where the protected class is not binary, but can take on multiple values. Here there are multiple challenges including defining an appropriate version of fairness.

**Acknowledgments**

Flavio Chierichetti was supported in part by the ERC Starting Grant DMAP 680153, by a Google Focused Research Award, and by the SIR Grant RBSI14Q743.

## Footnotes

[1]Given a graph with edges costs and capacities, a source, a sink, the goal is to push a given amount of flow from source to sink, respecting flow conservation at nodes, capacity constraints on the edges, at the least possible cost.

[2]https://archive.ics.uci.edu/ml/datasets/diabetes

[3]https://archive.ics.uci.edu/ml/datasets/Bank+Marketing

[4]https://archive.ics.uci.edu/ml/datasets/adult

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
