[Reviews · NeurIPS 2017]

Reviewer 1



While there is a growing body of work on fair supervised learning, here the authors present a first exploration of fair unsupervised learning by considering fair clustering. Each data point is labeled either red or blue, then an optimal k-clustering is sought which respects a given fairness criterion specified by a minimum threshold level of balance in each cluster. Balance lies in [0,1] with higher values corresponding to more equal numbers of red and blue points in every cluster. This is an interesting and natural problem formulation - though a more explicit example of exactly how this might be useful in practice would be helpful. For both k-center and k-median versions of the problem, it is neatly shown that fair clustering may be reduced to first finding a good 'fairlet' decomposition and then solving the usual clustering problem on the centers of each fairlet. While this is NP-hard, efficient approximation algorithms are provided based on combining earlier methods with an elegant min cost flow construction. On 3 example datasets, it is shown that standard clustering yields low balance (perhaps not surprising). A preliminary investigation of the cost of fairness is provided where balance of at least 0.5 is required and the number of clusters is varied. It would be interesting to see also the sensitivity to balance: keep the number of clusters fixed, and vary the required balance. Minor points: l. 16 machine learning are -> machine learning is In a few places, e.g. l. 19, \citep{} would improve the readability, so Kleinberg et al. (2017a) -> (Kleinberg et al. 2017a). l. 53 define -> defines. In fact, I think you just look at the costs of regular clusterings compared to fair so don't really use this ratio? Fig 1 caption, would give -> might give. x is assigned to a -> x is assigned to the same cluster as a l. 102 other measure -> other measures, please could you elaborate? l. 132, 136 Y -> Y_j as in l. 161 Near Definition 5, an image showing an example of how fairlets together make up the fair clustering would be helpful, though I appreciate it's hard given space constraints - in the Supplement would still be good.

Reviewer 2



The paper studies the problem of performing fair clustering under the disparate impact doctrine as definition of fairness for binary protected classes, e.g., the gender. The paper introduces a measure of fairness in clustering, which is named as balance; and introduce the concept of fairlets as the minimal sets satisfying fairness while preserving the clustering objectives. With these two notions in the hand, the authors state the problem of learning fair clustering as a two step problem, where the fist stage aims to find the fairlets and the second cluster (as in standard clustering algorithms) these fairlets into clusters. However, the authors prove that finding fairlets is an NP-hard problem under both k-center and k-median clustering objectives that, however, this problem can be approximated in polynomial time. The technical contributions of the paper are significant, and the paper in general is readable (although mathematically heavy at times). In general, I believe that the paper constitutes one of the first attempts to deal with algorithmic fairness in the context of unsupervised learning, which is an open and very interesting problem from the community. It also shows that there is still a lot of future work to do in this area: First, balance as fairness measure is only applicable to balanced datasets in terms of sensitive attributes values. Second, the proposed approaches are only valid for one binary protected classes like the gender. Third, the solution of the algorithm is still approximate and the bounds proved in the paper still need to be tighten. The main room for improvement in the paper is the experimental section, where the authors made little effort to validate and analyze the results of the proposed algorithms. In particular, an increase in the size of Figure 3 (it is hardly readable) as well as further explanation and analysis of its results would improve the quality of the paper.

Reviewer 3



In this paper the authors address the generation of 'fair' clusters. The authors acknowledge there is no clear or unique definition of fairness, they decided to use the notion of disparate impact. Based on this, they derived versions of well-known clustering algorithms, also providing some theoretical guarantees and an algorithmic properties. The paper is well-written and organised. It is in general easy to follow. The related work section could be improved though. It is not always clear what the connection is between this work and others (especially with Kemel 2013). Also, a couple more papers could be relevant for the discussion. https://ai2-s2-pdfs.s3.amazonaws.com/08a4/fa5caead14285131f6863b6cd692540ea59a.pdf https://arxiv.org/pdf/1511.00830.pdf They also address fair clustering algorithms, so I find the statement '... we initiate the study of fair clustering algorithms' too strong. The notion of fairlets I find interesting and appealing. The theoretical analysis seems sound. The experimental section is the weakest part of the paper. As it is, it doesn't add much to the story. A comparison with at least a realistic baseline is needed.